# The "Law" of the Unconscious Contrastive Learner: Probabilistic Alignment of Unpaired Modalities

**Yongwei Che**
Princeton University
yongweic@princeton.edu

**Benjamin Eysenbach**
Princeton University
eysenbach@princeton.edu

## Abstract

While internet-scale data often comes in pairs (e.g., audio/image, image/text), we often want to perform inferences over modalities unseen together in the training data (e.g., audio/text). Empirically, this can often be addressed by learning multiple contrastive embedding spaces between existing modality pairs, implicitly hoping that unseen modality pairs will end up being aligned. This theoretical paper proves that this hope is well founded, under certain assumptions. Starting with the proper Bayesian approach of integrating out intermediate modalities, we show that directly comparing the representations of data from unpaired modalities can recover the same likelihood ratio. Our analysis builds on prior work on the geometry and probabilistic interpretation of contrastive representations, showing how these representations can answer many of the same inferences as probabilistic graphical models. Our analysis suggests two new ways of using contrastive representations: in settings with pre-trained contrastive models, and for handling language ambiguity in reinforcement learning. Our numerical experiments study the importance of our assumptions and demonstrate these new applications.

Code: https://github.com/YongweiChe/UnconsciousContrastiveLearner

## 1 Introduction

Much of the appeal of contrastive learning is that it gives a "plug-n-play" approach for swapping one modality for another. Because representations from different modalities are trained to be aligned when representing the same object, the hope is that (say) a language representation and image representation of the same scene can be used as substitutes. This property is practically appealing for at least two reasons. *First*, it allows us to make use of pre-trained models. If you have a model that wants to make use of (say) language input and you have access to a pre-trained image-language contrastive model, you might simply train your model on the pre-trained image representations and hope that it will continue to work when you swap in the language representations. *Second*, it allows you to merge datasets that have different modalities, in the same way that a data scientist might merge two SQL tables. This is especially useful in settings where we have datasets with *pairs* of modalities, where collecting datasets with *triplets* of modalities (e.g., image, audio, and text) is challenging. While it seems intuitive that this "plug-n-play" approach should work, to the best of our knowledge prior work has not provided a rigorous argument for why it works or when it should fail. The main contribution of our paper is to answer these questions, and to provide an alternative algorithm for when this "naive" approach fails.

Our analysis is based on two ideas. The *first idea*, building on prior work (Poole et al., 2019; Ma & Collins, 2018), is to think about contrastive learning probabilistically. Rather than thinking about representations as encoding "this image maps to this text," we think about them as encoding a likelihood – that this text is more likely to describe this image, but could plausibly also describe this other image. The *second idea*, building on top of other prior work (Wang & Isola, 2020), is to consider the marginal distribution over the representations. We follow prior works in making the assumption that the marginal distribution is either *(1)* Gaussian and *(2)* Uniform over the hypersphere.

The main contribution of this paper is a probabilistic analysis of multimodal contrastive learning. The *first part* of our analysis is to show that, when certain assumptions are satisfied, the "plug-n-play" approach is principled, correctly inferring the posterior mean. We also introduce an approach

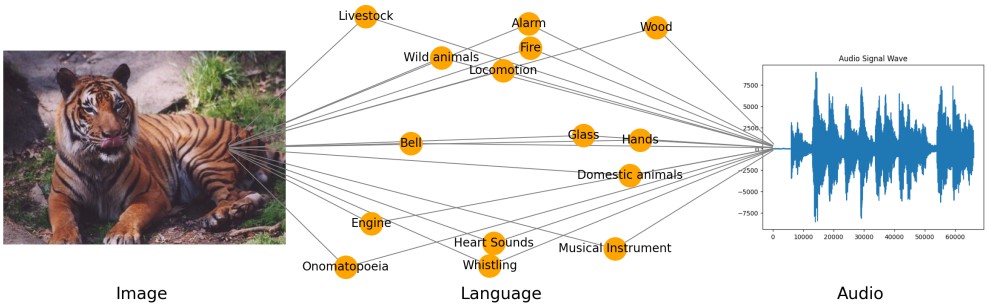

Figure 1: Aligning vision and audio by integrating over the intermediate 'language' modality. While prior work has shown that aligning modalities $A \leftrightarrow B$ and $B \leftrightarrow C$ results in representations that can compare modalities A and C, it remains unclear when and why this is guaranteed to work. This paper provides the assumptions under which this approach is principled, and our analysis unlocks new ways of comparing unpaired modalities and new applications for contrastive learning.

that works agnostic to the marginal distribution of the representations. The *second part* looks at applications of our results. We start by verifying our theoretical results with experiments over synthetic datasets. One such experiment includes a setting where users want to bridge modalities $A$ and $C$ using pre-trained or black-box models that connect $A$ and $C$ through an intermediate modality $B$. Then, we extend our results to large pre-trained contrastive models over image, language, and audio. Finally, we explore an application of our method in language-conditioned reinforcement learning, testing our method's ability to handle natural ambiguity in language.

## 2 RELATED WORK

Our work builds on a wide range of prior work, ranging from representation learning, multimodal machine learning, and probabilistic graphical models. Contrastive learning methods aim to learn compact representations of examples from one or more modalities (Chen et al., 2020; Hadsell et al., 2006; Mikolov et al., 2013), and have found applications ranging from ecology to NLP to to reinforcement learning. These methods typically are trained on pairs of examples, with the aim of acquiring representations that are similar for paired examples and dissimilar for unpaired examples.[1] While these pairs are often generated via data augmentation (Oord et al., 2018), our work builds on prior work that looks at paired examples coming from different modalities (e.g., an image accompanied by its caption) (Elizalde et al., 2023; Girdhar et al., 2023; Zhu et al., 2023). Prior work has shown that these methods can work effectively with more than two modalities Xue et al. (2023); Guzhov et al. (2021), even in settings where there are no paired examples from certain modalities (Girdhar et al., 2023; Zhu et al., 2023; Shah et al., 2023; Yuan et al., 2023; Ferraro et al., 2023). However, from a theoretical perspective it remains unclear why these sorts of "plug-n-play" methods can combine modalities for which there are no training examples, a problem that ordinarily requires marginalization (which is very challenging on the high-dimensional inputs to which these methods have been applied). Are these methods implicitly performing this marginalization? Are they correctly accounting for uncertainty?

Our paper will aim to answer some of these questions by building on top of ideas from two lines of prior work. First, we will exploit the geometry of contrastive representations (Wang & Isola, 2020; Eysenbach et al., 2024), borrowing (theoretically-motivated) assumptions about the asymptotic behavior of contrastive representation distributions. Second, we build on prior work (Poole et al., 2019) that provides a probabilistic interpretation of contrastive learning: when trained with a cross entropy loss, the learned representations encode a probability ratio. When combined, these two ideas allow us to prove that reasoning across unpaired modalities takes the form of a certain integral over representations, which admits a closed form solution. Our theoretical analysis will explain why the "plug-n-play" approach often works, elucidate its assumptions, and provide new ways of combining unpaired modalities that work under fewer assumptions.

---

[1]Our work will not consider the successful line of prior work that studies negative-free contrastive learning methods (Grill et al., 2020; Chen et al., 2021).

## 3 PRELIMINARIES

This section formally defines the problem we are studying: performing probabilistic inferences across pairs of modalities that do not simultaneously occur in the training data.

### 3.1 PROBLEM STATEMENT

Formally, assume there exists three data modalities $A, B, C$, and we have large-scale paired datasets for $A \leftrightarrow B$ and $B \leftrightarrow C$, but little to no data of $A \leftrightarrow C$. Our goal is to learn three encoders $\phi_A(A), \phi_B(B), \phi_C(C)$ that map each modality into a $d$-dimensional latent representation.

Our focus will be on applying contrastive methods to train these three encoders. Prior works jointly train encoders $\phi_A \leftrightarrow \phi_B$ and encoders $\phi_B \leftrightarrow \phi_C$ under contrastive losses Zhu et al. (2023); Girdhar et al. (2023). Using this training objective, these works achieve strong results in diverse applications. However, these works take the emergent alignment between unpaired modalities for granted. The alignment result is trivial if there exists a one-to-one mapping over all modality triplets $(A, B, C)$, however, it is unclear whether the method maintains the correct probabilistic information between the unpaired data modalities.

### 3.2 CONTRASTIVE LEARNING

Given two modalities $A$ and $B$, let $p(a, b)$ be the joint distribution over $a \in A$ and $b \in B$. We will learn a *critic function* $f$ that assigns a high score to "aligned" pairs $f(a, b) \sim p(a, b)$ and a low score to "unaligned" pairs $f(a, b) \sim p(a)p(b)$. This critic function is typically some distance between learned representations. We will use $\phi_A(\cdot) : A \to \mathbb{R}^d$ to denote the encoder that maps input $a$ to a $d-$dimensional representation, and likewise for $\phi_B(\cdot)$ and $\phi_C(\cdot)$. We will learn these encoders by comparing the similarity between the produced representations using a function $f(\phi_A(a), \phi_B(b))$.

We train these encoders by sampling pairs $(a, b^+) \sim p(a, b)$ of aligned examples (e.g., an image and its corresponding caption). We will also sample "un-aligned" pairs from the product of the marginal distributions: $(a, b^-) \sim p(a) \cdot p(b)$. Both these aligned and unaligned examples are used to compute the symmetrized infoNCE objective (Radford et al., 2021), which we optimize to train our encoders:

$$\min_{\phi_A, \phi_B} \sum_{i=1}^{N} \left[ \log\left( \frac{e^{f(a_i, b_i)}}{\sum_{j=1}^{N} e^{f(a_i, b_j)}} \right) + \log\left( \frac{e^{f(a_i, b_i)}}{\sum_{j=1}^{N} e^{f(a_j, b_i)}} \right) \right],$$

where the batch of training examples $(a_i, b_i)_{i=1}^{N}$ sampled from the joint distribution $p(a, b)$.

### 3.3 KEY ASSUMPTIONS

We outline two key assumptions necessary for our analysis, followed by a third assumption on the marginal distribution of our representations. The first assumption is a restriction on the relationships of the data modalities for our analysis to hold. The remaining assumptions build off prior work in contrastive learning.

**Assumption 1.** *Desired modalities $A$ and $C$ are conditionally independent given the intermediate "bridge" modality $B$. Formally, $A \perp C \mid B$*

While this assumption may appear limiting, it proves necessary for meaningful analysis on the relationship between $(A, C)$ within our problem setting. That is, given three random variables $A, B, C$, and oracle access to any probabilities involving $(A, B)$ and $(B, C)$ separately, but not to any probabilities involving all three variables together, it is generally impossible to uniquely determine $P(C|A)$ without additional assumptions. For a full proof, refer to Lemma 4 in the Appendix.

**Assumption 2.** *Applying contrastive learning to the symmetrized InfoNCE objective results in representations that model a density ratio: $e^{f(\phi_A(A), \phi_B(B))} = \frac{p(B|A)}{K \cdot p(B)}$.*

We denote $\phi_M$ as the data encoder for a modality $M$, $f(\cdot, \cdot)$ as a function that maps $d$-dimensional latent representations to a real number, $f : \mathbb{R}^d \times \mathbb{R}^d \to \mathbb{R}$, and $K$ is a constant representing the approximation error. This assumption is well justified by prior work (Poole et al., 2019; Ma & Collins,

2018), which proves that contrastive representations converge to this probability ratio under some assumptions. Nonetheless, we state this as an "assumption" to clarify that it could be violated in practice (e.g., if data is limited).

The last assumption is needed in just part of our analysis, when understanding the settings where the "plug-n-play" heuristic used by prior work is formally justified. The assumption is that the representations learned by contrastive learning have a certain marginal distribution.

**Assumption 3.** *Contrastive learning under symmetrized InfoNCE with an inner product $f(x, y) \triangleq x^\top y$ and unit-norm latent vectors acquires representations whose marginal distribution $p(\phi) \triangleq \int p(x)\mathbb{1}\{\phi_M(x) = \phi\}dx$ is uniform over the hypersphere $\mathbb{S}^{d-1}$: $p(\phi_M) = \mathcal{U}(\mathbb{S}^{d-1})$.*

Prior work (Wang & Isola, 2020) provides a theoretical basis for why this assumption should hold. We also test whether Assumption 3 holds in practice. See Section 6.2.2.

## 4 BAYESIAN REASONING OVER CONTRASTIVE REPRESENTATIONS

We show that representations learned jointly under InfoNCE correctly infer density ratios. Assumptions 1 and 2 will allow us to evaluate the direct connection between two unpaired modalities by using the Bayesian approach of integrating over the intermediate modality. Then, under Assumption 3, we can use the marginal distribution of the intermediate representation to reach a closed-form solution for the relationship between our disparate modalities.

### 4.1 MARGINALIZATION WITH CONTRASTIVE REPRESENTATIONS

The first part of our analysis shows how to evaluate the correct marginal density ratio between unpaired data modalities $A$ and $C$.

**Lemma 1.** *Let $\phi_A(a), \phi_B(b), \phi_C(c)$ be three encoders trained with contrastive learning on paired data $p(a, b)$ and $p(b, c)$. Assume that the encoder pairs $(\phi_A(a), \phi_B(b))$ and $(\phi_B(b), \phi_C(c))$ satisfy Assumption 2, and that our modalities $A, B, C$ satisfy Assumption 1. The marginal density ratio between $A$ and $C$ can be expressed as*

$$\frac{p(C \mid A)}{p(C)} = K_1 \cdot K_2 \cdot \mathbb{E}_{\phi_B}\left[\exp\{f(\phi_A, \phi_B) + f(\phi_B, \phi_C)\}\right]$$

*where $K_1, K_2$ respectively denote the constant multiplicative errors of $(\phi_A, \phi_B), (\phi_B, \phi_C)$ in approximating the marginal density ratio from Assumption 2.*

This result holds for any choice of $f(\phi_1, \phi_2)$ that is consistent with Assumption 2. This expression is what a proper Bayesian would do, if she were not allowed to make additional assumptions (i.e., Assumption 3). Note that the expression is reminiscent of doing message passing (Yedidia et al., 2003) on a graphical model $A \leftrightarrow B \leftrightarrow C$, except where the node potentials tell us about probability *ratios* instead of probabilities.

*Proof.* The proof follows by using Assumptions 1 and 2 together with Bayes' Rule:

$$\frac{p(C \mid A)}{p(C)} = \int_B \frac{p(C \mid B)}{p(C)} \frac{p(B \mid A)}{p(B)} \cdot p(B)\, dB$$

$$= \int_B K_1 \cdot e^{f(\phi_C, \phi_B)} \cdot K_2 \cdot e^{f(\phi_B, \phi_A)} \cdot p(B)\, dB = K_1 \cdot K_2 \cdot \mathbb{E}_{\phi_B}\left[e^{f(\phi_C, \phi_B) + f(\phi_B, \phi_A)}\right].$$

$\square$

Sec. 5 describes a practical and efficient approach for approximating this expectation with Monte Carlo samples. It is worth noting that Lemma 1 is *not* what is used today; this expression is different from directly comparing the representations $\phi(A)$ and $\phi(C)$, a baseline that we will compare against and discuss at length in the subsequent sections. However, in the following section, we will show that a version of the direct dot-product comparison is theoretically justified under additional assumptions. This will enable us to obtain an expression for the probability ratio without the need for an expectation. For subsequent derivations, we assume a constant approximation error and omit the $K_1 \cdot K_2$ term for clarity.

## 4.2 BUILDING INTUITION VIA THE TRIANGLE INEQUALITY

Before presenting our main result, we attempt to build intuition for why directly comparing the representations of $\phi(A)$ and $\phi(C)$ would ever be a good idea. We will do this in the setting where the representations are normalized, and measure the representation similarity as $f(\phi_1, \phi_1) = \phi_1^\top \phi_2$; our subsequent results will also examine different choices. With this choice of critic, we can get a *lower bound* on the log ratio:

$$\log \frac{p(C \mid A)}{p(C)} = \log \mathbb{E}_{\phi(B)} \left[ e^{\phi(A)^\top \phi(B) + \phi(B)^\top \phi(C)} \right] \geq \log \mathbb{E}_{\phi(B)} \left[ e^{\phi(A)^\top \phi(C)} \right] = \phi(A)^\top \phi(C).$$

The first inequality follows from the triangle inequality (note that the vectors are normalized). We removed the expectation in the third expression because the term inside the expectation does not depend on $\phi(B)$.

While this identity provides some intuition into why there should be a connection between the inner products and the log ratio, it is not the identity that we want: we want to estimate the log ratio itself, not a lower bound on it. Additionally, while the analysis above is tight in the setting where $\phi(A)$ and $\phi(C)$ are orthogonal (akin to the Pythagorean theorem), this would imply that $A$ and $C$ are independent, contradicting our intuition that modality $A$ should tell us something about modality $C$. Thus, we will need a different proof technique for our main result.

## 4.3 THE "LAW" OF THE UNCONSCIOUS CONTRASTIVE LEARNER

This section provides the main theoretical result of the paper, showing that a commonly-used heuristic is justified under some assumptions. Prior work often assumes that the dot product between representations from unpaired modalities meaningfully describes their similarity, somehow marginalizing over any uncertainty. Our proof here provides some theoretical justification for that claim, alongside the necessary assumptions.

**Lemma 2** (The "Law" of the Unconscious Contrastive Learner). *Let $\phi_A(a), \phi_B(b), \phi_C(c)$ be three encoders trained with contrastive learning on paired data $p(a, b)$ and $p(b, c)$ to output normalized representations (i.e., $\|\phi\|_2^2 = 1$) using $f(\phi_1, \phi_2) = \phi_1^\top \phi_2$. Assume that the underlying joint distribution $p(a, b, c)$ satisfies Assumption 1. Assume that the encoder pairs $(\phi_A(a), \phi_B(b))$ and $(\phi_B(b), \phi_C(c))$ satisfy Assumption 2, and assume that all three encoders satisfy Assumption 3.*

*Then, the probability ratio is a deterministic function of the inner product, $\frac{p(C|A)}{p(C)} = g(\phi_A(A)^\top \phi_C(C))$, where $g(x) = \frac{(2\pi)^{p/2} I_{p/2-1}(x)}{\|x\|_2^{p/2-1}}$ is a **monotonically increasing** function of $x$ defined in terms of a modified Bessel function $I_v$.*

This Lemma is important because it provides a theoretical grounding for the commonly used heuristic of comparing representational similarity between modalities unseen together during training.

*Proof.* Before starting the proof, we note that it will make use of the von-Mises-Fisher distribution, a probability density function over the unit hypersphere that has density $f(x; \mu, \kappa) = C_p(\kappa) \exp(\kappa \mu^\top x)$ where the normalizing constant $C_p(\kappa) = \frac{\kappa^{p/2-1}}{(2\pi)^{p/2} I_{p/2-1}(\kappa)}$ is defined in terms $I_p$, the modified Bessel function of the first kind.

Our proof starts by applying Lemma 1 with $f(\phi_1, \phi_2) = \phi_1^\top \phi_2$, and then proceeds by using Assumption 3 to write the marginal $p(\phi(B))$:

$$\frac{p(C \mid A)}{p(C)} = \mathbb{E}_{\phi(B)} \left[ e^{\phi(A)^\top \phi(B) + \phi(B)^\top \phi(C)} \right]$$

$$= \int_{\mathbb{S}^{d-1}} e^{\phi(B)^\top (\phi(A) + \phi(C))} \, d\phi_B$$

$$= \int_{\mathbb{S}^{d-1}} e^{\kappa \mu^\top \phi(B)} \, d\phi_B \qquad \text{where } \kappa = \|\phi(A) + \phi(C)\|_2 \text{ and } \mu = \frac{\phi(A) + \phi(C)}{\|(\phi(A) + \phi(C))\|_2}$$

$$= \frac{1}{C_p(\kappa)} \underbrace{\int_{\mathbb{S}^{d-1}} C_p(\kappa) e^{\kappa \mu^\top \phi(B)} \, d\phi_B}_{1}.$$

To conclude, we note that $\kappa$ can be written in terms of the inner product between $\phi_A(a)^\top \phi_B(b)$ as the representations norms equal 1: $\frac{1}{C_p(\|\phi(A)+\phi(C)\|)} = \frac{1}{C_p(\sqrt{2+\phi(A)^\top \phi(B)})} = g(\phi(A)^\top \phi(B))$.

$\square$

### 4.4 EXTENSION TO UNNORMALIZED REPRESENTATIONS

In Appendix B, we extend this analysis to the case of unnormalized representations with the following Lemma. Here we will assume that the representation distribution is Gaussian, rather than uniform (as in Assumption 3); see Wang & Isola (2020); Eysenbach et al. (2024) for a justification of this assumption.

**Lemma 3.** *Consider applying contrastive learning with the symmetric infoNCE loss and encoder $\phi_A, \phi_B, \phi_C$ on paired data $p(A, B)$ and $p(B, C)$, using a critic function parametrized as $f(x, y) \triangleq -\frac{1}{2}\|x - y\|_2^2$. Following prior work, assume that the acquired representations have a marginal distribution that is an isotropic Gaussian: $p(\phi) \triangleq \int p(x) \mathbb{1}\{\phi_M(x) = \phi\}dx = \mathcal{N}(\phi_M; \mu = 0, \sigma = c \cdot I)$. Then, the learned encoders $\phi_A$ and $\phi_C$ encode the true probability ratio $\frac{p(C|A)}{p(C)}$, despite never being trained on pairs of $(A, C)$ data:*

$$\frac{p(C \mid A)}{p(C)} = K \cdot \exp\{-\gamma \left(\|\phi(A) - \phi(C)\|_2^2 + \delta \phi(A)^\top \phi(C)\right)\}.$$

*where $\gamma = \frac{c+1}{4c+2}, \delta = \frac{1}{c+1}$.*

Note that as $c \to \infty$, we have $\delta \to 0$. The proof can be found in Appendix B.

In Lemmas 2 and 3, we recover closed-form expressions for the marginal probability ratio of modalities $A$ and $C$ with just their representations $\phi(A)$ and $\phi(C)$ respectively. For the dot product critic, the log probability ratio can be computed as a monotonically increasing transform of the dot product between $\phi(A)$ and $\phi(C)$; and for the negative $l_2$ distance critic, the log probability ratio asymptotically approaches the negative $l_2$ distance between $\phi(A)$ and $\phi(C)$ as $c$ increases. We see that the relationship between $\phi(A)$ and $\phi(C)$ for both normalized and unnormalized representations **closely resemble** their respective critic functions! We refer to this phenomenon as the "Law" of the Unconscious Contrastive Learner.

## 5 A PRACTICAL ALGORITHM FOR WHEN THE "LAW" DOES NOT HOLD.

In this section we turn Lemma 1 into a practical algorithm for estimating the log ratio $\frac{p(C|A)}{p(C)}$ given inputs $A$ and $C$. We will form a Monte Carlo approximation of the expectation in Lemma 1:

$$\frac{p(C \mid A)}{p(C)} \approx K \cdot \left(\frac{1}{N} \cdot \sum_{i=1}^{N} \exp\{f(\phi_A, \phi_i) + f(\phi_i, \phi_C)\}\right),$$

where $\phi_i \sim p(\phi_B(B))$ is the representation of a randomly sampled example of modality $B$. The approximation here reflects sampling error for the Monte Carlo summation and function approximation error from contrastive learning. In the limit of infinite samples ($N$) and contrastive representations trained to convergence on infinite data, the approximation becomes exact. Note that this result requires Assumptions $1 - 2$, **but not Assumption 3**.

In settings where we want to compute this probability ratio for many different values of $A$ and $C$, we can save compute by precomputing a matrix of representations $\Phi = [\phi(B), B \sim p(B)] \in \mathbb{R}^{N \times d}$:

$$\frac{p(C \mid A)}{p(C)} = K \cdot \left(\frac{1}{N} \cdot \sum_{i=1}^{N} \exp\{f(\phi_A, \Phi_i) + f(\Phi_i, \phi_C)\}\right).$$

This expression then can be computed very efficiently in modern deep learning libraries (PyTorch, JAX) as a LogSumExp. We study the efficacy of this method in Sec. 6 and show how it can enable new applications in Sections 6.2 and 6.3. For brevity, we will refer to this method of Monte Carlo approximation as the **LogSumExp (LSE)** method.

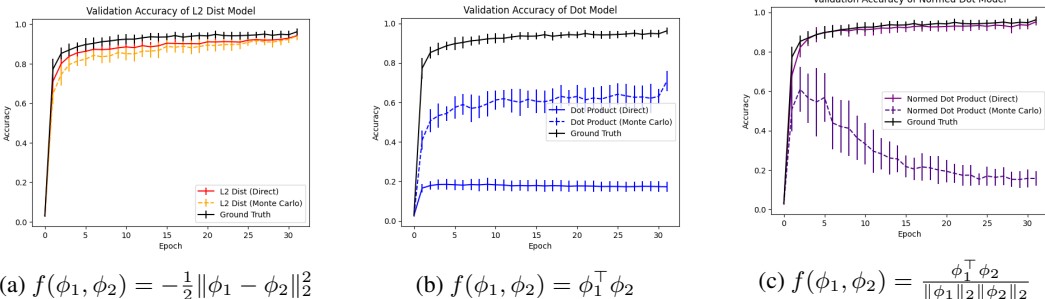

(a) $f(\phi_1, \phi_2) = -\frac{1}{2}\|\phi_1 - \phi_2\|_2^2$    (b) $f(\phi_1, \phi_2) = \phi_1^\top \phi_2$    (c) $f(\phi_1, \phi_2) = \frac{\phi_1^\top \phi_2}{\|\phi_1\|_2 \|\phi_2\|_2}$

Figure 2: **Testing the "Law" of the Unconscious Contrastive Learning** with three parametrizations of the critic function. We assess whether the "Law" holds by comparing the success of the "Direct" method to an oracle that is trained on $(A, C)$ examples. We also include the Monte Carlo method based on Lemma 1 to understand the assumptions belying our analysis. *(Left)* For the L2 critic, all methods perform well, suggesting that all three assumptions are satisfied. *(Center)* For the dot product critic, the Direct method performs much worse than the Monte Carlo method, suggesting that Assumption 3 is violated. Fig. 7 confirms that Assumption 3 is violated. *(Center)* The performance of the Direct and Monte Carlo methods when using the normalized dot product suggests that the normalized dot product violates Assumption 2, but that this assumption might not be necessary for the "Law" to hold.

## 6 EXPERIMENTS

The primary aim of our experiments is to validate our theoretical results and understand how those results depend on the aforementioned assumptions. We also demonstrate how Lemma 1 opens up new avenues for using contrastive representations: for combining multiple pre-trained models, and for correctly handling ambiguity in a language-conditioned reinforcement learning problem.

We start by describing the experimental setup we use for most of the experiments. Appendix Fig. 8 runs an additional experiment studying the influence of Assumption 1. Code for all experiments is available on GitHub.

### 6.1 EXPERIMENTS ON SYNTHETIC DATA

We first test our contrastive models on a synthetic dataset of modalities $A, B, C$. We generate $B$ from a $n_B$-dimensional Gaussian distribution parameterized by a random Covariance matrix $\Sigma$. $A$ and $C$ are generated from $B$ through a random linear projection ($M_A \in \mathbb{R}^{n_A \times n_B}, M_B \in \mathbb{R}^{n_C \times n_B}$) as well as additional uncorrelated Gaussian noise ($\epsilon_A, \epsilon_B$).

$$B \sim \mathcal{N}(\mu; \Sigma), \qquad A := M_A B + \epsilon_A, \qquad C := M_B B + \epsilon_B.$$

We generate two separate paired datasets of $(A, B)$ and $(B, C)$ to train our models. We evaluate the model's performance by measuring its retrieval accuracy: given a data point $a \in A$ we assess whether the model can accurately identify the corresponding data point from a list of 32 candidate data points in $C$, using maximum similarity score. We compute similarity scores using the model's critic function. We present 1-sigma error-bar plots to show the mean and standard deviation of the validation accuracy over many trials, which is computed every epoch.

We train on datasets with 5000 pairs and evaluate accuracy on a validation dataset of size 1000. For all synthetic experiments, we define accuracy as the Recall@1 metric: for each query item, we rank all candidate items by their similarity scores and check if the correct match appears at rank 1. We run the experiment for 20 trials with random seeds for $\mu, \Sigma, M_A, M_B$.

#### 6.1.1 TESTING THE "LAW" OF THE UNCONSCIOUS CONTRASTIVE LEARNER

The aim of this section is to develop a better understanding of the assumptions behind the "Law." We will see the "Law" need not hold when the assumptions are violated.

We learn contrastive representations $\phi_A, \phi_B, \phi_C$ using three choices of similarity function $f(\phi_1, \phi_2)$. For each, we measure the retrieval accuracy throughout the course of learning. Our gold standard is a "Ground Truth" method that has (privileged) access to $(a, c)$ pairs, data that the other methods do not assume to have. The "Direct" method performs retrieval by comparing the representations $\phi_A$ and $\phi_C$, representations that were not trained together, using the similarity function; the success of

this method tells us about when the "Law" holds. We also compare with the "Monte Carlo" method suggested by Lemma 1. This method makes fewer assumptions, so its performance helps us figure out the role of different assumptions in the "Law."

We report results in Fig. 3. In Fig. 2a we see that all methods perform well, suggesting that the assumptions are satisfied. In Fig. 2b, we see a gap between the Monte Carlo method and the Direct method, suggesting that Assumption 3 is violated. While we do not provide formal proofs regarding the unnormalized dot product, we hypothesize that its representation distribution is highly skewed and fails to meet the sufficient conditions for our "Law". This hypothesis is supported by Fig. 7, which illustrates that the unnormalized representations (green) exhibit irregular behavior, violating a key assumption of our framework. In addition, the gap between the Monte Carlo method and the Ground Truth methods here suggests that there may be optimization challenges in satisfying Assumption 2 with the dot product critic. Finally, Fig. 2c shows that the Monte Carlo method performs poorly when using a normalized dot product. Intuitively, this makes sense, as this critic cannot represent log probabilities outside $[1/e, e]$.[2] However, the Direct method still achieves excellent results in the setting. Fig. 7 suggests that the representations do satisfy Assumption 3, but the reasoning above suggests they likely violate Assumption 2.

In summary, our results show that our theoretical analysis provides a *sufficient* set of conditions for the "Law" to hold, and suggest a remedy for settings where Assumption 3 is violated. However, the results in Fig. 2c suggest that these conditions may not always be necessary, opening the door to future work to find even less restrictive conditions under which the "Law" holds.

## 6.2 BRIDGING MODALITIES WITH PRETRAINED MODELS OR BLACK BOX APIS

In many practical applications, users only have access to the similarity scores between pairs of modalities, but nonetheless want to reason about the similarities of unpaired modalities. For example, let's say a user wants to compute the similarity between $A \leftrightarrow C$, but searching the internet they find one pretrained contrastive model for $A \leftrightarrow B$ and (e.g., in some other GitHub repository) another pre-trained contrastive model for $B \leftrightarrow C$. These contrastive models may have different architectures and their representations may have different sizes. As another example, imagine that there are two ML web services, one that returns the similarity score between modalities $A$ and $B$ (e.g., a radiology service that aligns X-rays with text) and a second that computes the similarity score between modalities $B$ and $C$ (e.g., a dictation service that aligns text with speech). The underlying models are proprietary, so the user cannot inspect their weights.

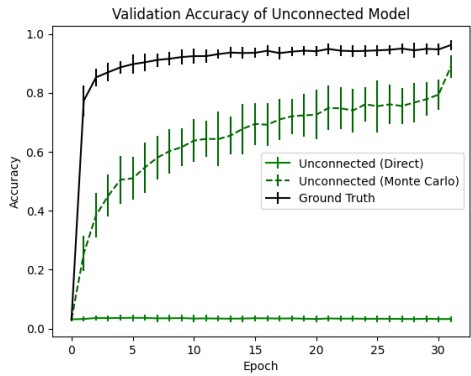

Figure 3: A principled way of combining pre-trained models. Given pre-trained models that compute the similarities $A \leftrightarrow B$ and $B \leftrightarrow C$, we use Lemma 1 to infer the similarity between $A \leftrightarrow C$.

The approach introduced in Sec. 5 handles both these problem settings. We first evaluate this approach on the synthetic benchmark described in Sec. 6.1, measuring performance by retrieval accuracy. We train contrastive critic $\phi_A \leftrightarrow \phi_{B_1}$ and $\phi_{B_2} \leftrightarrow \phi_C$, using the $l_2$ critic for both. We show results in Fig. 3. As expected, direct evaluation between representations $\phi_A$ and $\phi_C$ performs no better than chance. However, supporting our theory, we see that the Monte Carlo method achieves an accuracy rate approaching that of an oracle that is trained on aligned $(a, c)$ pairs.

### 6.2.1 APPLICATION: LARGE PRE-TRAINED LANGUAGE MODELS

We corroborate the above results with experiments on **real-world data**, confirming our theoretical results. We apply our methodology to three models—CLIP, a large pre-trained vision-language model Radford et al. (2021); CLAP, a large pre-trained audio-language model Elizalde et al. (2023); and LanguageBind, a massively multimodal pre-trained contrastive model connecting modalities

---

[2]Including a temperature term in the critic expands the range of values, but the range remains finite.

(vision, audio, infrared, depth, etc.) via a shared language modality Zhu et al. (2023). Using these three models, we evaluate the performance of zero-shot Audio-Visual alignment. First, we assess CLIP/CLAP encoders to demonstrate the effectiveness of our Monte Carlo algorithm. Subsequently, we evaluate LanguageBind encoders to validate our proposed "Law".

We begin by showing how our Monte Carlo Method (LogSumExp) over pre-trained CLIP and CLAP models unlocks Audio-Visual inference with no additional training required. For direct evaluation, we measure the similarity score as the normalized dot product between the CLIP image encoder with the CLAP audio encoder, which is expected to do no better than random chance. We then apply our LogSumExp algorithm to the intermediate 'language' modality, using the CLIP and CLAP language encoders to connect the originally disparate Image and Audio encoders. We evaluate these models on AudioSet Gemmeke et al. (2017) – a dataset of short clips from YouTube with corresponding Audio descriptions. To perform our Monte Carlo (LogSumExp) method over the intermediate 'language' modality, we approximate the true distribution of intermediate language embeddings using the AudioSet ontology, which defines a universe of language descriptions that the clips fall into.

Given just the language ontology and pre-trained CLIP and CLAP models, we are able to achieve 62% Recall@10 using our Monte Carlo method. This beats the naive baseline of directly evaluating the similarity scores between the image embedding of CLIP and the audio embedding of CLAP, which achieves 14% Recall@10 accuracy. We also compare

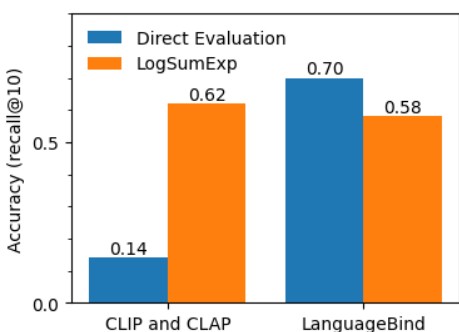

Figure 4: *(Left)* Direct evaluation of the CLIP Image Encoder with the CLAP Audio Encoder for audio-visual inference versus using our LogSum-Exp algorithm with those same encoders. *(Right)* Direct evaluation with LanguageBind encoders vs our LogSumExp algorithm with the those same encoders. In Appendix Fig. 5 we show that the 12% gap between the Directe Evaluation and LogSum-Exp on LanguageBind is caused by using too few Monte Carlo samples; this gap shrinks to zero as the number of Monte Carlo samples is increased, in line with our theory.

the performance of our LogSumExp algorithm against LanguageBind, which *implicitly* assumes our "Law" in its architecture. LanguageBind trains multiple encoders contrastively through an intermediate language modality, achieving superior results on many multi-modal benchmarks. Direct evaluation with LanguageBind achieves a 70% recall on the AudioSet benchmark, which further empirically validates our "Law". Our Monte Carlo approximation on the LanguageBind encoders achieves 58% recall. However, by scaling the intermediate embeddings beyond just the AudioSet ontology, we can close this performance gap (see Appendix C.1).

Our Monte Carlo approximation on CLIP/CLAP, while achieving lower accuracy than LanguageBind (62% vs 70% Recall@10), offers significant advantages. Notably, it enables the connection of previously disjoint models without additional training, demonstrating its flexibility and efficiency. Furthermore, our method requires data only from the intermediate modality, in contrast to prior approaches that demand data from all three modalities for full alignment Wang et al. (2023). This reduction in data requirements represents a substantial practical benefit, potentially broadening the applicability of multi-modal inference across diverse domains where comprehensive datasets are scarce.

### 6.2.2 Testing Representation Uniformity in Complex Data Modalities

We additionally validate that Assumption 3 fares well in complex real-world settings. We compute the representations of language descriptions for each model over the AudioSet ontology and perform a two-sample Kolmogorov-Smirnov test, comparing the distribution over representations to a uniform hypersphere distribution. CLIP (p-value = 0.0877) and CLAP (p-value = 0.1788) both fail to show significant deviation from the uniform hyperspherical distribution.

### 6.3 Application: Language-Conditioned Reinforcement Learning

Our analysis not only helps explain the zero-shot capabilities of contrastive learning methods but also enables novel applications in reinforcement learning. We demonstrate this through language-conditioned navigation tasks, where learned representations can be highly non-uniform and non-Gaussian, limiting direct evaluation effectiveness. Our Monte Carlo LogSumExp method significantly

improves performance in these scenarios, boosting success rates by 20%-30% across different environments.

We design a synthetic benchmark using a continuous PointMaze environment to evaluate language understanding in navigation. The environment consists of a grid where rows and columns are annotated with natural language descriptions (e.g., "the first column," "the second-to-last row"). The agent can move freely at any angle and must plan paths to reach locations specified through potentially ambiguous language descriptions $\ell$. We frame this as a reinforcement learning problem where an agent observes state $s_t$, takes action $a_t$, and transitions to state $s_{t+1}$. The objective is to select actions that reach the language-described destination in minimal steps.

To map this onto our multimodal framework from Sec. 3, we treat modality $A$ as state-action pairs $(s, a)$, modality $B$ as future states $s_f$ sampled several steps ahead, and modality $C$ as language descriptions $\ell$ of those future states. We learn encoders by aligning $\phi_A(s, a) \leftrightarrow \phi_B(s_f)$ and $\phi_B(s) \leftrightarrow \phi_C(\ell)$. The first alignment corresponds to contrastive reinforcement learning (Eysenbach et al., 2023), which learns policies by maximizing $\phi_A(s, a)^\top \phi_B(g)$ between state-action representations and goal state representations.

**Baseline.** Our primary comparison is against the direct representation comparison approach widely used in robotics and RL (Lynch et al., 2023; Sontakke et al., 2024; Tziafas et al., 2023), often utilizing CLIP embeddings (Radford et al., 2021). This baseline selects actions as $\max_a \phi_A(s, a)^\top \phi_C(\ell)$. *These methods implicitly employ the law of the unconscious contrastive learner*, and our experiments test whether this assumption is appropriate when uncertainty quantification matters.

**Our approach.** Following Lemma 1 and Sec. 5, instead of directly comparing state-action and language representations, we marginalize over future states:

$$\max_a \log \sum_{s_f \sim p(s_f)} e^{\phi_A(s,a)^\top \phi_B(s_f) + \phi_B(s_f)^\top \phi_C(\ell)}.$$

This directly maximizes the probability that future states satisfy the desired language description.

Our method demonstrates superior performance across three challenging environments (see Appendix D for detailed results). A key illustrative example occurs in our fork maze environment (Figure 9), where agents must navigate to "the first column." The direct method's primary failure mode is its tendency to move toward the mean position of ambiguous descriptions rather than finding optimal paths. For instance, when starting from position $(2, 6)$, direct evaluation takes a suboptimal rightward path before eventually reaching the goal through a different alleyway. In contrast, our LogSumExp method correctly identifies and follows the optimal leftward trajectory.

This performance disparity stems from the LogSumExp method's ability to maintain the full distribution over possible goal states rather than reducing ambiguous language to averaged embeddings. By summing over all intermediate states, our approach can identify actions that maximize the probability of reaching valid goals, even under ambiguous language specifications. For comprehensive analysis of the learned representations and additional experimental results, we refer readers to Appendix D.

## 7 CONCLUSION

This paper has two aims: to understand how to use contrastive representations to correctly reason about uncertainty, and to figure out when and why the commonly-used "direct comparison" heuristic works effectively. Our results provide a principled method for reasoning across unpaired modalities, along with analysis suggesting that the heuristic *is* principled, under some assumptions.

**Limitations.** The main limitation of our paper is that it does not provide the final word on when users should prefer the "direct comparison" approach over the Monte Carlo approach. In future work, we would like to design a simple test that users can apply to their data to understand whether it satisfies the assumptions, and guide users in selecting an algorithm for reasoning across modalities.

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

## A  AN IMPOSSIBILITY RESULT

**Lemma 4.** *Given three random variables $A, B, C$, and oracle access to any probabilities involving $(A, B)$ and $(B, C)$ separately, but not to any probabilities involving all three variables together, it is generally impossible to uniquely determine $P(C|A)$ without additional assumptions.*

*Proof.* We know that

$$p(C|A) = \int_B p(C|A, B) \cdot p(B|A).$$

However, both $p(B|A)$ and $p(C|A, B)$ depend on the unknown joint distribution $p(A, B, C)$. Knowing $p(A, B)$ and $p(B, C)$ is insufficient to uniquely determine $p(A, B, C)$, as there exist multiple joint distributions can share the same marginals $p(A, B)$ and $p(B, C)$ but differ in their dependence structure between $A$ and $C$ given $B$.

For example, let $A$, $B$, and $C$ be binary random variables with given joint distributions $p(A, B)$ and $p(B, C)$ that are uniform over all possible pairs.



Table 1: Joint distribution $p(A, B)$

| $p(A, B)$ | $B = 0$ | $B = 1$ |
|---|---|---|
| $A = 0$ | 0.25 | 0.25 |
| $A = 1$ | 0.25 | 0.25 |

Table 2: Joint distribution $p(B, C)$

| $p(B, C)$ | $C = 0$ | $C = 1$ |
|---|---|---|
| $B = 0$ | 0.25 | 0.25 |
| $B = 1$ | 0.25 | 0.25 |



We can construct two different joint distributions consistent with these marginals: one where $A$ and $C$ are independent given $B$ (yielding $p(C|A) = 0.5$ for all $A$), and another where $A$ and $C$ are perfectly correlated (so $p(C|A) = 1$ when $C = A$). Despite both joint distributions matching the given $p(A, B)$ and $p(B, C)$, they result in different $p(C|A)$, demonstrating that $p(C|A)$ is not uniquely determined without additional information. $\qquad\square$

## B  LAW OF THE UNCONSCIOUS CONTRASTIVE LEARNING (GAUSSIAN SETTING)

In the main text, we focused on the setting where representations lay on the unit hypersphere. Lemma 3 extended the analysis to *unnormalized* representations, whose distribution we assume to be a Gaussian, an assumption that is theoretically motivated based on prior work (Wang & Isola, 2020; Eysenbach et al., 2024). Here, we provide a proof of that Lemma.

*Proof.* We evaluate the integral under $\phi(B) \sim \mathcal{N}(0, c \cdot I)$. We start with the expectation:

$$\frac{p(C|A)}{p(C)} = \mathbb{E}_{\phi(B)} \left[ e^{-\frac{1}{2}\left( \|\phi(C)-\phi(B)\|_2^2 + \|\phi(B)-\phi(A)\|_2^2 \right)} \right] \tag{1}$$

$$= e^{-\frac{1}{2}\left( \|\phi(A)\|_2^2 + \|\phi(C)\|_2^2 \right)} \mathbb{E}_{\phi(B)} \left[ e^{-\left( \|\phi(B)\|^2 - \phi(B)^\top (\phi(A)+\phi(C)) \right)} \right] \tag{2}$$

Because $\phi(B)$ is Gaussian, we know $p(\phi(B)) = \frac{1}{(2\pi c)^{k/2}} e^{-\frac{1}{2c}\|\phi(B)\|^2}$. Thus,

$$\frac{p(C|A)}{p(C)} = e^{-\frac{1}{2}\left( \|\phi(A)\|^2 + \|\phi(C)\|^2 \right)} \cdot \frac{1}{(2\pi c)^{k/2}} \int_{\mathbb{R}^k} e^{-\left( \|\phi(B)\|^2 - \phi(B)^\top (\phi(A)+\phi(C)) + \frac{1}{2c}\|\phi(B)\|^2 \right)} d\phi(B) \tag{3}$$

$$= e^{-\frac{1}{2}\left( \|\phi(A)\|^2 + \|\phi(C)\|^2 \right)} \cdot \frac{1}{(2\pi c)^{k/2}} \int_{\mathbb{R}^k} e^{-\left( \left(1+\frac{1}{2c}\right)\|\phi(B)\|^2 - \phi(B)^\top (\phi(A)+\phi(C)) \right)} d\phi(B) \tag{4}$$

Letting $\alpha = 1 + \frac{1}{2c}, \quad y = \phi(A) + \phi(C)$, we complete the square to get

$$\frac{p(C|A)}{p(C)} = \frac{1}{(2\pi c)^{k/2}} \cdot e^{-\frac{1}{2}\left( \|\phi(A)\|^2 + \|\phi(C)\|^2 \right)} \cdot e^{\frac{\|y\|^2}{4\alpha}} \int_{\mathbb{R}^k} e^{-\alpha \left\| \phi(B) - \frac{y}{2\alpha} \right\|^2} d\phi(B) \tag{5}$$

$$= \frac{1}{(2\pi c)^{k/2}} \cdot e^{-\frac{1}{2}\left( \|\phi(A)\|^2 + \|\phi(C)\|^2 \right)} \cdot e^{\frac{\|y\|^2}{4\alpha}} \cdot \left( \frac{\pi}{\alpha} \right)^{k/2} \tag{6}$$

$$= \left( \frac{1}{(2\pi c)^{k/2}} \cdot \left( \frac{\pi}{\alpha} \right) \right)^{k/2} \cdot e^{-\frac{1}{2}\left( \|\phi(A)\|^2 + \|\phi(C)\|^2 \right) + \frac{1}{4\alpha}\left( \|\phi(A)\|^2 + 2\phi(A)^\top \phi(C) + \|\phi(C)\|^2 \right)} \tag{7}$$

$$= K \cdot \exp \left\{ -\left( \frac{c+1}{4c+2} \right) \|\phi(A) - \phi(C)\|^2 - \left( \frac{1}{2c+1} \right) \phi(A)^\top \phi(C) \right\} \tag{8}$$

Simplifying further, we finally arrive at

$$\frac{p(C \mid A)}{p(C)} = K \cdot \exp\{ -\gamma \left( \|\phi(A) - \phi(C)\|_2^2 + \delta \phi(A)^\top \phi(C) \right) \}.$$

where $\gamma = \frac{c+1}{4c+2}, \delta = \frac{2}{c+1}$. $\qquad\square$

## C  ADDITIONAL EXPERIMENTS

### C.1  SCALING UP THE MONTE CARLO APPROXIMATION FOR LANGUAGEBIND AND IMAGEBIND

In Section 6.2.1, we evaluated our Monte Carlo approximation against direct computation using LanguageBind on AudioSet data, observing a 12% performance gap. Given that our Monte Carlo

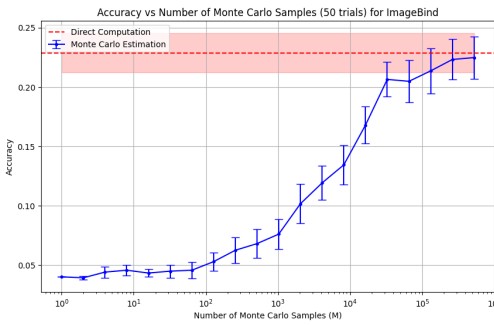 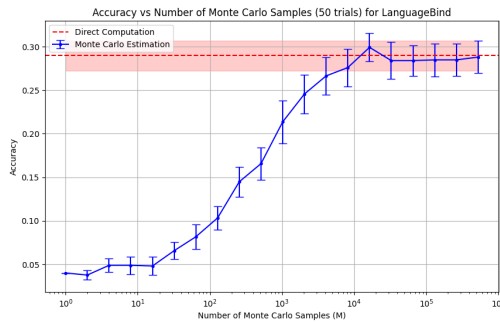

(a) Recall@1 Accuracy for ImageBind  (b) Recall@1 Accuracy for LanguageBind

Figure 5: The Accuracy of our LogSumExp (Monte Carlo) approximation scales with the number of intermediate embeddings. As the number of sampled embeddings (M) increases, the Monte Carlo method converges to direct evaluation performance with both ImageBind (left) and LanguageBind (right). The shaded regions indicate 95% confidence intervals across multiple trials, and the dashed red lines represent the accuracy of direct computation. Recall@1 is evaluated from a set of 25 samples. These results validate our theoretical analysis and support the underlying assumptions of our approach.

method relies on sampling intermediate embeddings, and the AudioSet ontology provides only 600-700 samples for this intermediate space, we investigate whether this gap stems from insufficient sampling or rather theoretical limitations.

We conduct a systematic scaling analysis using audio, image, and text modality triples $(A, B, C)$ from YouTube videos provided by AudioSet. We evaluate on two state-of-the-art multimodal models: LanguageBind, which aligns modalities through shared language descriptions, and ImageBind, which aligns modalities through shared image spaces.

Figure 5 demonstrates that as we increase the number of intermediate samples $(M)$ from 1 to 500,000, our Monte Carlo approximation converges to direct evaluation performance for both models. This convergence across different multimodal models validates our hypothesis that the initial performance gap was primarily due to limited sampling rather than a limitation of our method.

## C.2 SWAPPING MODALITIES WITHIN THE SAME SETUP

To further validate our theoretical findings, we conduct additional real-world experiments by varying the intermediate modality while maintaining the same experimental framework from Section 6.2.1.

### C.2.1 EXPERIMENTAL SETUP

Recall that our original setup in Section 6.2.1 leverages CLIP (Image-Language) and CLAP (Audio-Language) to align image and audio modalities through their shared language encoder. We design two additional experimental configurations:

1) **Image-Language Alignment via Audio**: We utilize ImageBind Girdhar et al. (2023) as our Image-Audio model paired with CLAP as our Audio-Language model. ImageBind provides high-quality audio-visual representations trained with InfoNCE loss, aligning with our theoretical assumptions. We evaluate our Monte Carlo algorithm against baselines from ImageBind (Direct Evaluation using ImageBind's Vision and Language encoders) as well as CLIP.

2) **Audio-Language Alignment via Images**: We utilize ImageBind as our Image-Audio model paired with CLIP as our Image-Language model. We evaluate our Monte Carlo algorithm against baselines from ImageBind (Direct Evaluation using ImageBind's Audio and Language encoders) as well as CLAP.

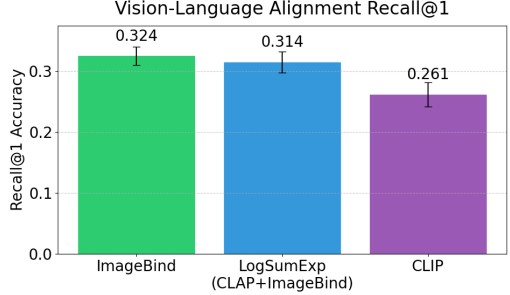
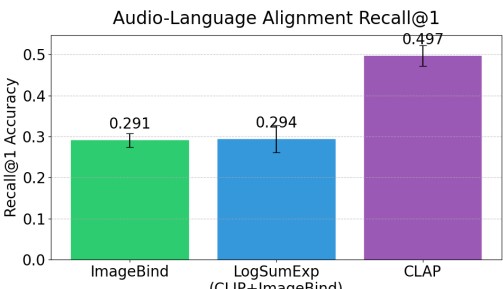

(a) Recall@1 Accuracy for Aligning Vision and Language.

(b) Recall@1 Accuracy for Aligning Audio and Language

Figure 6: LogSumExp performs well with other intermediate modality types (audio and vision). The error bars indicate 95% confidence intervals across 100 trials. Recall@1 is evaluated from a set of 25 samples. These results strengthen our real-world experiment in Section 6.2.1 and further validate our theoretical analysis.

### C.2.2 RESULTS AND ANALYSIS

For both Vision-Language and Audio-Language tasks, we compare our LogSumExp method against ImageBind's direct evaluation, which implicitly implements the "Law". We find that LogSumExp accuracy closely matches direct evaluation:

- In Vision-Language alignment, LogSumExp achieves a Recall@1 of $31.4\% \pm 1.7\%$, compared to ImageBind's direct evaluation performance of $32.4\% \pm 1.5\%$
- For Audio-Language alignment, LogSumExp reaches $29.4\% \pm 3.3\%$ versus ImageBind's $29.1\% \pm 1.7\%$

Baselines against Single-Model Baselines (CLIP and CLAP in purple) are provided in Figure 6. The near-identical performance between LogSumExp and Direct Evaluation validates that similar phenomena to Section 6.2.1 hold regardless of intermediate modality type. This provides additional empirical support for our theoretical framework, suggesting that our Monte Carlo algorithm generalizes across different choices of intermediate representations.

### C.3 REPRESENTATION DISTRIBUTION IN $\mathbb{R}^2$

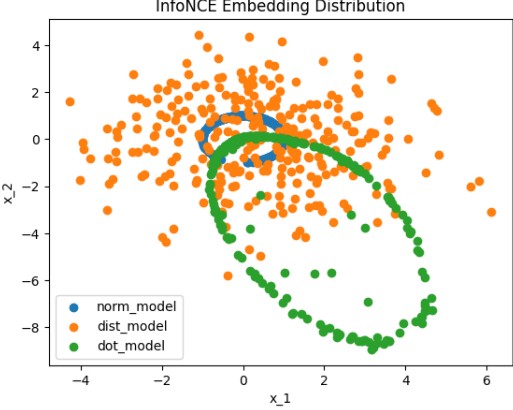

Figure 7: Embedding Distributions under different critic functions

For a subset of synthetic data experiments, we run with a latent embedding size of 2, which allows us to visualize the resulting representation distributions. See Figure 7 for the plots.

## C.4 Ablation on Conditional Independence

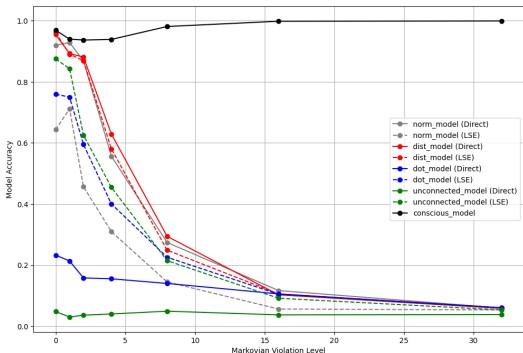

Figure 8: Performance degrades when Assumption 1 is violated. The x-axis corresponds to the magnitude of $\kappa$.

Assumption 1 requires the conditional independence of $A$ and $C$ given modality $B$. What happens when this assumption is violated? With the separating intermediate modality, it becomes impossible to observe the correlation between $A, C$. When this happens, we do not have $p(C \mid A, B) = p(C|B)$ anymore, so our analysis becomes approximate.

We construct a synthetic dataset for this case, building off the existing synthetic dataset in Section 6. We now add a Gaussian random variable $\kappa \in \mathbb{R}^1$ to modality pairs $A$ and $C$ that is unseen by $B$. Thus, $A, C$ are serially correlated independent of $B$.

$$B \sim \mathcal{N}(\mu; \Sigma) \tag{9}$$

$$A := M_A B + \epsilon_A + \kappa \vec{1} \tag{10}$$

$$C := M_B B + \epsilon_B + \kappa \vec{1} \tag{11}$$

We denote $\vec{1}$ as the all-ones vector. When $\kappa = 0$, our conditional independence assumption is satisfied. As $\kappa$ grows, our conditional independence of $A, C$ given $B$ decreases.

See Figure 8 for the accuracy plots. The $y$-axis denotes the retrieval accuracy of the methods and the $x$-axis denotes the magnitude of our correlation variable $\kappa$, described in the previous section. We see that the accuracy of our models unanimously goes down as our constant disturbance increases.

## D AN APPLICATION TO CONTRASTIVE REINFORCEMENT LEARNING

We delve briefly into contrastive reinforcement learning. Contrastive representations lend themselves very well to self-supervised reinforcement learning due to their inherent encoding of probability distributions over future states. Interestingly, for the same reason, contrastive learning also parsimoniously encodes uncertainty between image and text. Thus, if we are able to marry these two methods together under our framework, we will have a method for language goal-conditioned reinforcement learning. This is the setting where we specify a language goal of a future state, and try to learn an algorithm that can reach that specified language state without any explicit rewards.

### D.1 THE ENVIRONMENT

We define a continuous grid environment for the player to move around in. The player can move around freely at any angle, and is trying to reach a pre-specified goal state. We annotate the grid environment with natural language descriptions of grid positions. Examples include each row and column in the grid being given a street and avenue name respectively. The language labels are such that many of them are ambiguous, sometimes referring only to certain rows or blocks. This is to test the compositionality of the learned language embeddings and how they incorporate uncertainty.

## D.2 TRAINING METHODOLOGY

Building on prior work, we can frame reinforcement learning purely as a contrastive learning problem. We learn embeddings for $(s, a)$ and $(s_f)$ pairs, where

$$s := \text{current state}, a := \text{action}, s_f := \text{future state}$$

We assume a dataset of oracle trajectories is collected, giving $(s, a), s_f$ pairs to learn over. We also assume a dataset of states with language annotations. Then using our unconscious method, we can simultaneously learn an embedding between $s_f$ and $l :=$ language description.

Learning the $(s, a)$ embedding allows us to choose the best action for the agent by taking an $\arg\max$ over a set of candidate actions. In practice, we find that this leads agents down a deterministic path that could lead to cycles (an infinite loop). Thus, we sample actions probabilistically, weighted by the exponentiated dot product between the goal embedding and the $(s, a)$ embedding.

This application is especially beautiful because contrastive representations allow us to capture inherent *uncertainty* in language, and these representations can then be directly used downstream for the control task, due to the fact that both use contrastively-learned embeddings as the latent space.

Because the reinforcement learning algorithm only requires estimation of the marginal probability ratio, we can use pre-trained representations for both contrastive tasks.

## D.3 CONNECTION TO EXISTING LANGUAGE-CONDITIONED REINFORCEMENT LEARNING METHODS

Our proposed "Law" bears significant resemblance to techniques used in prior works for language-conditioned tasks. The majority of existing research in language-conditioned reinforcement learning approaches the problem as a modality-bridging challenge, aiming to connect textual-visual alignment with goal-conditioned reinforcement learning. Our "Law" provides a principled probabilistic perspective that formalizes and justifies best practices for integrating these two sub-tasks.

Text2Control Cachet et al. (2024) is one example of this approach. The algorithm decomposes the problem into two stages:

1. Text-to-goal generation
2. Goal-reaching

This work employs a Vision-Language Model (VLM) to translate a textual goal into a goal 'configuration' image that can then be fed into a goal-reaching policy. However, this decomposition introduces a critical flaw: it may lead to the selection of 'impossible' configurations given the initial state. This is a severe limitation that can significantly impair the system's performance and reliability.

In a separate line of work, Ahn et al. (2022) introduces a different approach. This method computes feasibility ('affordance') scores for future states conditioned on the current state. These feasibility scores are then combined with 'Instruction Relevance' scores to determine the agent's policy.

Our LogSumExp method bears striking similarities to Ahn et al. (2022), as we implicitly compute:

- Affordance: $p(\text{future state} \mid \text{current state})$
- Instruction Relevance: $p(\text{language description} \mid \text{future state})$

Our method thus provides a unified probabilistic framework that encapsulates and formalizes the intuitions behind these existing approaches. By doing so, it offers a more rigorous foundation for language-conditioned reinforcement learning, potentially leading to more robust and generalizable algorithms. Furthermore, our approach inherently addresses the limitation of potentially selecting 'impossible' configurations, as seen in Text2Control, by considering the affordance of future states.

## D.4 RESULTS

Our Monte Carlo method outperforms direct evaluation on all tested maze environments. A subset of the results are visualized in Figure 13.



Figure 9: Fork Maze Environment with multiple possible paths to the "first column"

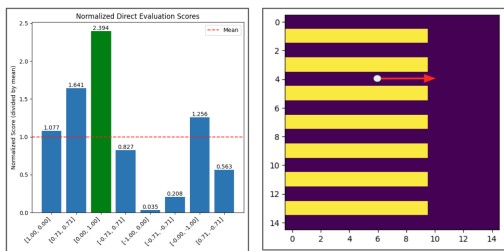

Figure 10: Direct evaluation chooses a suboptimal rightward action when given eight possible directions (unit vectors), failing to find the shortest path to the "first column."

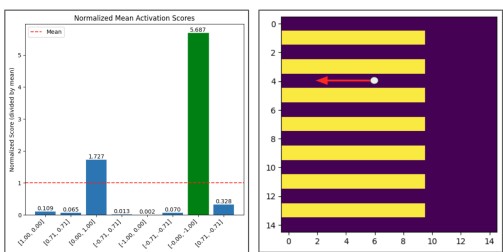

Figure 11: LogSumExp correctly identifies moving left as the optimal action for the same navigation task.

### D.5 AN AMBIGUOUS LANGUAGE EXAMPLE

To demonstrate the advantages of our Monte Carlo LogSumExp approach in handling ambiguous language navigation, we present an illustrative example in a $14 \times 14$ maze environment. The maze features a fork-like pattern of walls across its left half, with each row forming a separate alleyway (Figure 9).

We first train encoders $\phi_{A,B}$ on state-action pairs and future states $(s,a) \leftrightarrow (s_f)$, and encoders $\phi_{B,C}$ on states and language descriptions $s \leftrightarrow l$. We then task an agent at position $(2,6)$ to navigate towards the "first column." While the shortest path is directly left, direct evaluation fails to find this optimal route. As shown in Figure 10, an agent using direct navigation takes a significant detour, first moving right before eventually entering a different alleyway. This behavior reveals a fundamental limitation: direct evaluation cannot find the shortest path to a language-described destination, instead navigating towards the "mean" position of the ambiguous description. In contrast, Figure 11 shows that LogSumExp navigation correctly chooses to move left.

This performance difference stems from the state encoder's violation of Assumption 3. To understand LogSumExp's decision-making process, Figure 12 visualizes the critic functions for both the language description "the first column" (left) and various actions (right) against state embeddings from each grid cell. The $Q$-value for each action can be understood as the sum of compositions between the left language graph and the respective action graphs on the right. Formally, we compute:

$$\sum_{s_f \sim p(s_f)} e^{\phi_A(s,a)^\top \phi_B(s_f) + \phi_B(s_f)^\top \phi_C(\ell)}.$$

The "left" action achieves the highest $Q$-value because it aligns the most probable "next" states with the high-activation "language" states, leading the agent to choose this optimal direction.

This example illustrates a key advantage of our LogSumExp approach: its ability to handle ambiguous language descriptions by considering the full distribution over possible goal states. While direct evaluation reduces the language description "first column" to a single averaged embedding—leading to suboptimal navigation—our Monte Carlo LogSumExp method maintains the multimodal nature of the goal distribution. By summing over all possible intermediate states, it can identify that moving left maximizes the probability of reaching valid goal states, even when those goals are ambiguously

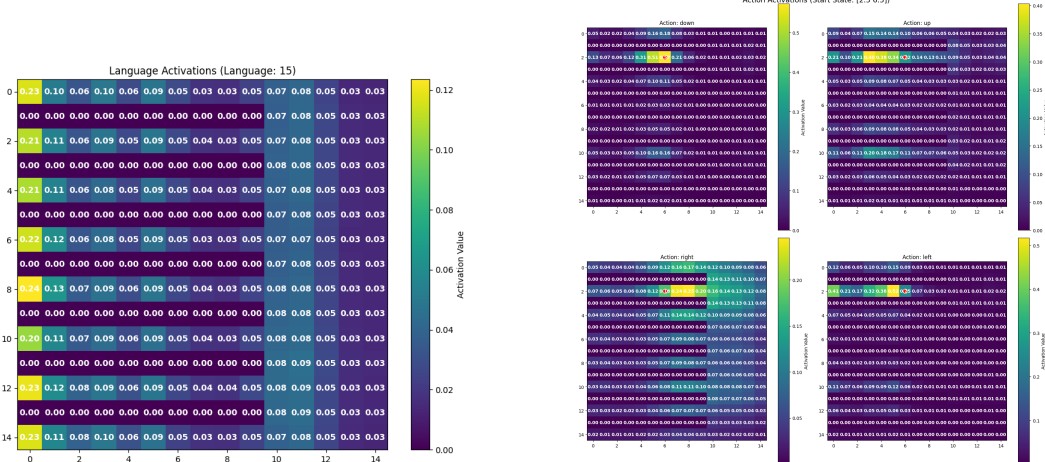

Figure 12: Monte Carlo approximation visualization over the intermediate state modality, sampling uniformly over valid grid points. Left: Average critic function evaluations for each cell with the language description "the first column". Right: Average critic function evaluations for each cell with the state-action pair (current state, action) for up, down, left, and right actions.

specified in language. This capability is particularly valuable in real-world applications where natural language commands often have multiple valid interpretations.

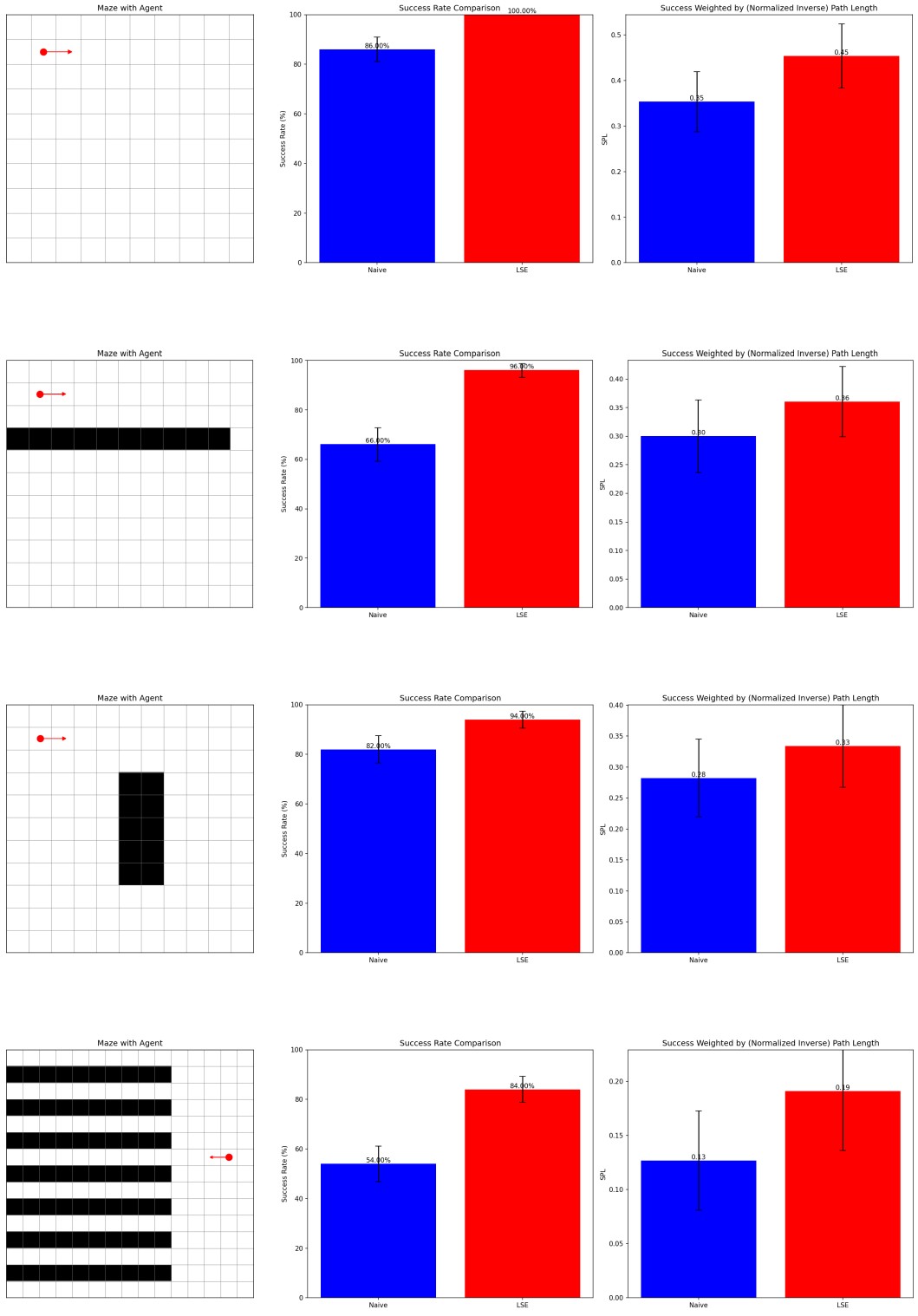

Figure 13: Success Rates of Direct Comparison vs LSE on different Grid Environments

