# OpenReview forum: "The "Law'' of the Unconscious Contrastive Learner: Probabilistic Alignment of Unpaired Modalities"
_ICLR.cc/2025/Conference — ICLR 2025 Poster_

### Official Review · Reviewer_RYrX · 2024-11-03

**Soundness:** 2
**Presentation:** 2
**Contribution:** 2
**Rating:** 5
**Confidence:** 3

**Summary:**

This paper tries to provide a theoretical justification to the scenario of using the third modality to connect two modalities which are not explicitly trained during contrastive learning. A typical use case is to conduct image-audio retrieval tasks through CLIP and CLAP with language as the intermediate.

Authors prove that such heuristic is theoretically grounded, by making a conditional independence assumption (of the two modalities to be
connected), along with two ideas published in previous work.

Two numerical experiments with synthetic and real-world datasets are employed to validate the effectiveness of the proof.

**Strengths:**

- It is desired to have some theoretical justifications to the common
  use case or practice of connecting two modalities that are not
  explicitly trained during contrastive learning. There are published
  work, but not much theoretical justifications contained.

**Weaknesses:**

- The method derived from the proof, LogSumExp, seems not working well for
  LanguageBind, and it leads to performance drop (Figure4).

- Overall the proposed proof seems mostly effective on synthetic
  datasets. I am thus concerned about how strong are the assumptions,
  especially assumption 1 -- are modalities A (e.g., audio) and C
  (e.g., image) really independent conditioned on B (language)?

- Writing needs to be improved; Sec. 6.3 experiment is difficult to
  understand with the main text alone.

**Questions:**

Line457. LanguageBind is claimed to assume the proposed Law
implicitly. However, when applying the law explicitly, why the
performance actually drops? I was expecting no impacts on LanguageBind
if the Law is implicitly implemented already.

---

> ### Author Response · Authors · 2024-11-22
> **Author Response**
>
> Dear Reviewer,
>
> Thanks for the detailed review and constructive feedback. To incorporate the reviewer's feedback and answer important questions, we have run two additional experiments.  The first new experiment shows that the previously observed Monte Carlo performance drop is primarily a function of sample size, and the performance drop rapidly diminishes as we increase the number of Monte Carlo samples (see Fig. 5). The second new experiment shows that our theoretical results also also apply to an additional real-world model (ImageBind) (see Fig. 5). **Together with the further revisions discussed below, do these new experiments fully address the reviewer's concerns?** We look forward to continuing the discussion, and would be happy to make additional revisions or run additional experiments.
>
> > Why does performance drop on LanguageBind
>
> Our new experiments demonstrate this gap is primarily due to limited Monte Carlo (MC) sampling. In Figure 5, we show:
> - With 1000 MC samples: 12% performance gap
> - With 100,000 MC samples: <3% gap
> - With 500,000 MC samples: <1% gap, matching direct computation This scaling analysis was validated across different models (LanguageBind, ImageBind), demonstrating consistent convergence behavior.
>
> > real-world data
>
> To study this question, we have run additional experiments with more models on real-world data (LanguageBind and ImageBind). The new results, found in Fig. 5, validate our Monte Carlo algorithm as well as Assumptions 1 and 2.
>
> > are modalities A (e.g., audio) and C (e.g., image) really independent conditioned on B (language)
>
> We agree that, in practice, this assumption may be violated (see Sec. 5). We have included an ablation experiment studying what happens when this assumption is violated in Appendix C.3. We believe that our theoretical results are important because they are the first (to the best of our knowledge) theoretical characterization of why the direct comparison method should work; this result is important because (1) it helps explain that this prior approach is not a heuristic, but (2) that it implicitly makes this assumption and so may fail in certain settings (see, e.g., Appendix C.3).
>
> > Section 6.3 is difficult to understand from the main text alone
>
> We have revised the exposition in Section 6.3 to improve clarity and better integrate its connection with the supporting materials in the Appendix. Does this sufficiently address the reviewer’s concerns? If not, we would be happy to further revise the paper.
> We would like to thank the reviewer again for the feedback! We believe the additional experiments and clarifications have provided additional support for the theoretical framework. Please let us know if we have addressed the concerns.
>
> Kind Regards,
>
> Tha Authors

---

> > ### Author Response · Authors · 2024-11-25
> > **Author Response**
> >
> > Dear Reviewer,
> >
> > We have worked hard to incorporate the review feedback by running new experiments and revising the paper. We'd really appreciate if you could confirm whether these changes address the concerns about the paper. If we have misunderstood any concerns we'd like to learn that now so we can further revise the paper or run additional experiments.
> >
> > Thank you!
> >
> > The Authors

---

> > > ### Comment · Reviewer_RYrX · 2024-11-26
> > > **Thank you for your response**
> > >
> > > Thanks for updating the results and the draft accordingly. I now understand that the performance gap is mostly due to limited MC samples, and the experiments on RL. I updated my score accordingly.
> > >
> > > In generaly I am still leaning negative, and the main concern is still on the applicability to real-world. The updated results are still on the same real-world dataset AudioSet (which is noisy itself), while the others remain on synthetic data. I feel more solid experiments on extra real-world and standard benchmakr datasets, like ImageNet-1K (for image-text retrieval) or AudioCaps/Clotho (for audio-text retrieval) would be helpful to better validate the proposed method.

---

> > > > ### Author Response · Authors · 2024-11-30
> > > > **Author Response**
> > > >
> > > > Dear Reviewer,
> > > >
> > > > Following the suggestion, we have conducted new audio-text retrieval experiments on the AudioCaps dataset. In addition, we have previously added Appendix C.2 that tests both audio-text and image-text retrieval on AudioSet by performing LogSumExp over different intermediate modalities (image and audio respectively). These new experiments further validate our method's real-world applicability across different modality combinations.
> > > >
> > > > The new AudioCaps experiments are run on its test set using the same experimental setup as Appendix C.2.1. Our evaluation measures Recall@1 accuracy on sets of 25 samples, with our Monte Carlo algorithm computing LogSumExp over sampled image frames from videos in the AudioCaps training set.
> > > >
> > > > Below are the audio-text retrieval results on AudioSet and AudioCaps:
> > > >
> > > > Dataset | Baseline | LogSumExp | Direct (ImageBind) | CLAP |
> > > > |---------|----------|-----------|-------------------|------|
> > > > | AudioSet | 0.040 | 0.294±0.035 | 0.291±0.020 | 0.497±0.040 |
> > > > | AudioCaps | 0.040 | 0.468±0.018 | 0.568±0.017 | 0.795±0.016 |
> > > >
> > > > The substantial improvement of LogSumExp over the baseline provides strong validation of our theoretical framework. These results further demonstrate that our method generalizes well to standard audio-text retrieval benchmarks beyond AudioSet. Do these new experiments fully address the reviewer's concerns?
> > > >
> > > > As the revision period has ended, we will include the new AudioCaps experiments in the camera-ready version of the paper.

---

> ### Comment · Area_Chair_SRzm · 2024-11-25
> **Are author responses satisfactory?**
>
> Have they improved their presentation and addressed other weaknesses?

---

### Official Review · Reviewer_Xnbz · 2024-11-03

**Soundness:** 3
**Presentation:** 3
**Contribution:** 3
**Rating:** 5
**Confidence:** 2

**Summary:**

Existing work often assumes alignment between different modality pairs without providing a theoretical basis. This paper, motivated by (1) isotropic representations and (2) a Bayesian probabilistic framework, introduces three assumptions that theoretically ensure alignment.

In a synthetic experiment, the authors first demonstrate that alignment fails when Assumption (iii)—that acquired representations have a marginal distribution following an isotropic Gaussian—is violated. They then show that retrieval accuracy remains high (indicating successful alignment) when Assumptions (i) and (ii), which involve Monte Carlo approximations, are satisfied, using an intermediate modality like language.

Finally, in a real-world application, the authors apply contrastive representations from pre-trained contrastive models within a probabilistic framework to manage language ambiguity in reinforcement learning.

**Strengths:**

1. This paper proposes theoretical assumption to guarantee the universally agreed alignment for multi-modalities.
2. The experiments on synthetic data verifies the correctness of the assumptions.
3. The real application experiment show promise of the assumption in reinforcement learning for addressing handling language ambiguity.

**Weaknesses:**

It is not clear why the proposed framework is better at language ambiguity situation, is it because that the framework is based on a probabilistic framework, i.e., the uncertainty.

The proposed assumptions seem to less contribute to the real-data application, i.e., language ambiguity in reinforcement learning framework, which limits its practical contribution.

**Questions:**

See weakness above.

1. Any practical implication based on the assumptions.
2. Detailed explanations of the reason why the framework can help language ambiguity, is that possible applied to other frameworks. The motivation of taking different multi-modality data into consideration in the RL settings.

---

> ### Author Response · Authors · 2024-11-22
> **Author Response**
>
> Dear Reviewer,
>
> Thank you for the detailed review and constructive feedback. It seems like the main suggestions relate to clarifying how our framework handles language ambiguity and studying its applicability to real-world applications. To incorporate these suggestions, we have added a step-by-step analysis of the language ambiguity experiment (Appendix D.5) and run new experiments on a separate real-world model (ImageBind) (See revised Fig. 5). Together with the discussion and further revisions discussed below, **do these new experiments and revisions fully address the reviewer's concerns?** If not, we would be happy to run additional experiments or further revise the paper.
>
> > Detailed Explanation of why LogSumExp can help language ambiguity
>
> We agree that the benefits to handling language ambiguity stem from our probabilistic framework. We have added a step-by-step analysis in Appendix D.5 to demonstrate these benefits.
>
> > The proposed assumptions seem to less contribute to the real-data application
>
> To study this question, we have run additional experiments with new models on real-world data (ImageBind and LanguageBind). The new results, found in Fig. 5, validate our Monte Carlo algorithm as well as Assumptions 1 and 2.
>
> We agree that theoretical results make certain assumptions, but our empirical results (see Fig. 4) have already shown the importance of these results in real-data settings where the assumptions may be violated. One contribution of our paper is to highlight that a heuristic that is already commonly used in practice ("direct comparison") implicitly relies on a certain assumption (Assumption 3). By highlighting this assumption, our paper (1) provides the first (to the best of our knowledge) proof of why this heuristic often works well in practice, (2) why this heuristic can fail in some settings, and (3) provides a new method (Monte Carlo) that continues to work when this assumption is violated (such as language-conditioned reinforcement learning).
>
> Thanks for raising these important questions about our theoretical and experimental results. The additional experiments and visualizations have strengthened our paper. Please let us know if we have addressed the concerns.
>
> Kind Regards,
>
> The Authors

---

> > ### Author Response · Authors · 2024-11-25
> > **Author Response**
> >
> > Dear Reviewer,
> >
> > We have worked hard to incorporate the review feedback by running new experiments and revising the paper. We'd really appreciate if you could confirm whether these changes address the concerns about the paper. If we have misunderstood any concerns we'd like to learn that now so we can further revise the paper or run additional experiments.
> >
> > Thank you!
> >
> > The Authors

---

> ### Comment · Area_Chair_SRzm · 2024-11-25
> **Are author responses satisfactory?**
>
> Rebuttals are coming to a close. Have the author clarifications addressed your concerns?

---

> ### Author Response · Authors · 2024-12-01
> **Author Response - Rebuttals Ending Soon**
>
> Dear Reviewer,
>
> We have worked hard to incorporate the review feedback by running new experiments and revising the paper. As we haven't yet received your response to our rebuttal and the discussion period is coming to an end, we would really appreciate if you could confirm whether our changes address your concerns about the paper.
>
> Kind Regards,
>
> The Authors

---

### Official Review · Reviewer_i8rh · 2024-11-04

**Soundness:** 3
**Presentation:** 3
**Contribution:** 3
**Rating:** 8
**Confidence:** 2

**Summary:**

This paper proves under certain assumption that the “hope” of unseen modality pairs will be aligned when the embedding space of models between existing modality pairs are trained contrastively. Using Bayesian approach, the paper shows that directly comparing the representation of data from unpaired modalities can recover the same likelihood ratio. The analysis shows that contrastive representations can answer many of the same inferences as probabilistic graphical models. The result of the paper suggests that the contrastive representations can be used in settings with pre-trained contrastive models, and for handling language ambiguity. Experiments are done to verify the theoretical results over synthetic datasets and more realistic setting.

**Strengths:**

- The three assumptions are clearly stated, and proofs as well as empirical tests for the assumptions are provided. The derivation of the conclusion that the relationship between the $\phi$ functions for both normalized and unnormalized representation closely resemble their critic functions.
- Empirical evaluations using the Monte Carlo approximation validates some assumptions’ validity, and also brings the possibility of only use intermediate modality data to align all the modalities’ representations. Experiments are done in both synthetic setup (section 6.1, 6.3) and real-world data (Section 6.2).

**Weaknesses:**

Section 6.1’s setup might be a bit artificial. Since modality A and C are only a projection of B, it does not quite suffice the general definition of “modality” that most of us agrees on.

**Questions:**

- In equation from 4.2, $\phi(A)^T\phi(B) + \phi(B)^T\phi(C) \geq \phi(A)^T\phi(C)$ is because the trained pairs are almost guaranteed to be closer than the unseen pair?
- Missing subscript $C$ for $\phi_C(C)$ in line 247?
- Apologize for not being familiar enough with the theory of the work, why is von-Mises-Fisher the distribution in Lemma 2? Is it just because the normalization constant or wanting to match assumption 3?
- The citation format in section 6.2.1 should be changed. E.g., AudioSet (Gemmeke et al., 2017).
- Monte Carlo approximation can be expensive. What is the cost in computation and time efficiency?

---

> ### Author Response · Authors · 2024-11-22
> **Author Response**
>
> Dear Reviewer,
> Thank you for your detailed review and constructive feedback. We especially appreciate the detailed review of the formal proofs in the paper. We have also incorporated the reviewer feedback into the revised paper (see below).
>
> > Intuition for Equation 4.2
>
> Equation 4.2 follows from the triangle inequality – note that all vectors are unit length. We have added this to the paper.
>
> > Why the von-Mises-Fisher distribution in Lemma 2?
>
> The reviewer is correct that we use the vMF distribution in Lemma 2 to match Assumption 3. Based on prior work, we assume our intermediate embedding distributions are uniformly distributed over the unit hypersphere. The von-Mises-Fisher distribution with parameter $\kappa = 0$ captures this uniform distribution.
>
> > Monte Carlo computation and time efficiency
>
> While Monte Carlo methods can be computationally expensive, in our case it only incurs a large one-time cost. The majority of computation time comes from generating embeddings for the intermediate representation (running inference for one image through ImageBind takes **23 milliseconds**). However, this needs to be done only once and can be cached for all future computations. Subsequently, the time complexity becomes linear in the number of Monte Carlo samples with a small constant factor (for an embedding dimension of 512, a single dot product takes **16 microseconds**).
>
> > Section 6.1 does not suffice as a modality
>
> We agree that Section 6.1 does not meet the conventional definition of a modality. We merely use this as a didactic example to illustrate our theoretical framework, before progressing to complex, real-world modalities in subsequent experiments in Sec. 6.2.
>
> > Typos
>
> We have fixed these in the revised paper.
>
> We would like to thank the reviewer again for their detailed comments regarding this work. Please let us know if there are any additional questions or concerns.
>
> Kind Regards,
>
> The Authors

---

> ### Comment · Reviewer_i8rh · 2024-11-25
> **Response to the authors**
>
> Thank you for your response, and it answers my questions :). As my score right now is pretty high, I will maintain my score.

---

### Official Review · Reviewer_rpJk · 2024-11-04

**Soundness:** 1
**Presentation:** 3
**Contribution:** 2
**Rating:** 5
**Confidence:** 3

**Summary:**

A common belief in contrastive multimodal learning is that the embedding spaces of seen modality pairs (A&B and B&C) naturally align with those of unseen modality pairs (A&C). This paper introduces the "Law of the Unconscious Contrastive Learner", showing that under specific assumptions about the geometric and probabilistic properties of contrastive embeddings, it is possible to establish relationships between unseen modality pairs. The law relies on three key assumptions: the first two allow for evaluating the connection between two unpaired modalities (A&C) via a Bayesian approach that integrates over an intermediate modality (B), while the third enables the use of the intermediate representation marginal distribution to derive a closed-form solution. The law is then formalized into a practical algorithm using Monte Carlo approximations. It is validated through experiments on synthetic and real datasets, including CLIP and CLAP. Results show the alignment of unpaired modalities, though some assumptions may not be strictly necessary in practice.

**Strengths:**

1. The idea of aligning representations of the same concept or sense, regardless of modality, is valid and taps into a widely-held belief. Although this is commonly accepted, it has not been well grounded in theory. This paper makes strides by validating this belief under certain assumptions. It provides a foundation that may guide further theoretical development in contrastive embedding alignments.
2. Although not checking line-by-line, the proposed theory assumes probabilistic contrastive learning, allowing connections between unpaired modalities (A&C) to be established across a shared intermediate modality (B) using Bayes Rule. Its algorithm using Monte Carlo approximations is backed by empirical evidence, which is interesting compared to the Oracle methods, where modality pairs (A&C) can be seen.
3. The paper is well-presented, especially the first half theoretical sections. This clarity makes the framework and assumptions easy to follow. Results shown in Figure 2 are helpful for a good understanding.

**Weaknesses:**

1. The experimental setup in Section 6 lacks standard organization, making it difficult to immediately grasp the task, input, output, and evaluation metrics. A clearer presentation of each experiment would improve readability and understanding.
2. There appears to be a gap between the Monte Carlo approximation method and the main theoretical framework. For example, Assumption 3 is noted as not strictly necessary, introducing ambiguity in the experiments and leaving some uncertainty about the necessity of all three proposed assumptions given the results.
3. Of the three main experiments, both the first (6.1) and third (6.3) are conducted on in-house datasets, and only the second experiment applies the algorithm to an external dataset (AudioSet) by comparing CLIP and CLAP, but its scope, datasets, model types, and modality pairings are limited.

**Questions:**

An easy way to strengthen the experimental findings could be to swap the modalities within the same setup. Theoretically, there is no strict assignment of text, image, or audio to roles A, B, or C, so these modalities could be switched to test the robustness of results.

(There appears to be a formatting issue in the proof of Lemma 2 on page 5.)

---

> ### Author Response · Authors · 2024-11-22
> **Author Response**
>
> Dear Reviewer,
>
> Thank you for your detailed review and your positive and constructive feedback. To address the concern about the the Monte Carlo approximation, we have run a new experiment (Fig. 5) showing that this gap shrinks towards zero as we increase the number of samples; note that our theoretical results only say that the methods should be the same in the limit of infinite samples (i.e., computing the exact expectation). We incorporate the other suggestions, we have made substantial revisions to the paper organization and added new results on another real-world model (ImageBind, in addition to the real-world experiments already included in the initial submission). Together with the discussion below, **do these new experiments and revisions fully address the reviewer's concerns?** If not, we'd be happy to run additional experiments and make further revisions to the paper.
>
> > additional real-world experiments
>
> We have added an experiment with ImageBind (see revised Fig. 5), which provides an additional modality pairing example (Aligning Audio and Text through the shared Image modality).
>
> > Gap in the Monte Carlo approximation method and the main framework
>
> The 12% gap in Fig 4 is caused by using a small number of Monte Carlo samples in the original paper. When we increase the number of samples from 600 → 500,000, we observe that this gap shrinks to almost zero. This is in line with our theoretical results, which say that these methods should be equivalent if the expectation is computed exactly. We have added a new Fig. 5 to show how this gap shrinks to zero as the number of samples is increased.
>
> > Assumption 3 is noted as not strictly necessary, introducing ambiguity in the experiments
>
> We have revised Sec. 5 to clarify that Assumption 3 is necessary for our direct comparison method (prior work) but not for the  Monte Carlo approach (our approach). Our experiments validate this distinction: direct evaluation and Monte Carlo perform well when uniformity holds (Section 6.2.2 and Figure 5), while the Monte Carlo algorithm excels on highly non-uniform reinforcement learning data (Section 6.3).
>
> One important contribution of our paper is to highlight that a heuristic that is already commonly used in practice ("direct comparison") implicitly relies on Assumption 3. By highlighting this assumption, our paper (1) provides the first (to the best of our knowledge) proof of why this heuristic often works well in practice, (2) why this heuristic can fail in some settings, and (3) provides a new method (Monte Carlo) that continues to work when this assumption is violated (such as language-conditioned reinforcement learning).
>
> > The experimental setup in Section 6 lacks standard organization
>
> We have revised the organization of Section 6.3 to clarify the connection between our theoretical framework and the RL experiments, while providing more rigorous empirical validation. We have also clarified the use evaluation metrics in section 6 for the synthetic data experiments.
>
> > Lemma 2 formatting issue
>
> We have made revisions to Lemma 2 and its proof.
>
> >  swap the modalities within the same setup.
>
> Thanks for this suggestion! We have started working on this by running experiments on using audio and image as the intermediate modality. We've been prioritizing the other requested experiments and haven't finished this experiment yet. We will include the final results in the camera ready version of the paper.
>
> Kind regards,
>
> The Authors

---

> > ### Author Response · Authors · 2024-11-25
> > **Author Response**
> >
> > Dear Reviewer,
> >
> > We have worked hard to incorporate the review feedback by running new experiments and revising the paper. We'd really appreciate if you could confirm whether these changes address the concerns about the paper. If we have misunderstood any concerns we'd like to learn that now so we can further revise the paper or run additional experiments.
> >
> > Thank you!
> >
> > The Authors

---

> > > ### Comment · Reviewer_rpJk · 2024-11-25
> > >
> > > Thanks for the response, which is helpful, particularly in clarifying Assumption 3 and its role in the work. It would be interesting to see if similar observations can be found in additional experiments after swapping modalities. Since I have not seen such experiments yet, I will maintain my position for now. That said, given the authors' commitment to addressing the feedback in the next version, I believe the updated work has the potential for acceptance, even if not here.

---

> > > > ### Author Response · Authors · 2024-11-26
> > > > **Author Response**
> > > >
> > > > Dear Reviewer,
> > > >
> > > > Thank you for your clear feedback. We have now added the requested experiments in Appendix C.2, where we test our framework using audio as an intermediate modality for vision-language alignment and vision as an intermediate modality for audio-language alignment. The results strongly support our theoretical analysis, showing that LogSumExp closely matches (within one percentage point) direct evaluation accuracy regardless of the intermediate modality type.
> > > >
> > > > In addition to these modality swapping experiments, we have already revised the paper and added new experiments to address the three weaknesses noted in your original review. We believe these revisions have strengthened the paper. Do these changes fully address your original concerns? If not, we would be happy to run additional experiments or further revise the paper.
> > > >
> > > > Kind regards,
> > > >
> > > > The Authors

---

> > > > > ### Author Response · Authors · 2024-12-01
> > > > > **Author Response - Rebuttals Ending Soon**
> > > > >
> > > > > Dear Reviewer,
> > > > >
> > > > > We have worked hard to incorporate the additional feedback by running the requested experiments and revising the paper (See Appendix C.2). We have also included additional audio-text retrieval experiments on the AudioCaps dataset [1]. We would really appreciate if you could confirm whether these changes address the concerns about the paper.
> > > > >
> > > > > Kind regards,
> > > > >
> > > > > The Authors
> > > > >
> > > > > -------------------
> > > > >
> > > > > [1] Additional AudioCaps Experiment Details:
> > > > >
> > > > > The experiments were run on the AudioCaps test set using the same setup as Appendix C.2.1. We measured Recall@1 accuracy on sets of 25 samples, with our Monte Carlo algorithm computing LogSumExp over sampled image frames from videos in the AudioCaps training set.
> > > > >
> > > > > Results for audio-text retrieval:
> > > > >
> > > > > Dataset   | Baseline | LogSumExp    | Direct (ImageBind) | CLAP
> > > > > ----------|----------|--------------|-------------------|-------------
> > > > > AudioSet  | 0.040    | 0.294±0.035  | 0.291±0.020      | 0.497±0.040
> > > > > AudioCaps | 0.040    | 0.468±0.018  | 0.568±0.017      | 0.795±0.016
> > > > >
> > > > > The substantial improvement of LogSumExp over the baseline provides strong validation of our theoretical framework. These results further demonstrate that our method generalizes well to standard audio-text retrieval benchmarks beyond AudioSet. As the revision period has ended, we will include the new AudioCaps experiments in the camera-ready version of the paper.

---

> ### Comment · Area_Chair_SRzm · 2024-11-25
> **Rebuttals ending soon, please discuss further.**
>
> Have the new experiments addressed your concerns? Is the presentation now satisfactory?

---

### Meta-Review · Area_Chair_SRzm · 2024-12-19

**Metareview:**

This paper presents a theoretical framework for understanding when multimodal knowledge alignment occurs. They prove three assumptions are key multimodal learning, all involving approximations to the distribution of the learned representations.

Pros:
- The multimodal framework is topical and their approaches principled in the theory.
- Their synthetic experiments confirm that when some assumptions are violated, models fail to align across modalities.
- They include several more natural multimodal datasets in the experiments.
- Their framework explains how ambiguity in language can be better handled through explicitly contrastive learning.

Cons:
- Skeptical of the connection between the theoretical and real-world setting. More realistic experiments were not finished during rebuttals, and the last semi realistic experiments are not strong on their own.
- Most experiments are very artificial or use noisy datasets.

**Additional Comments On Reviewer Discussion:**

Reviewer rpJk did not update their score because the new results were not yet added to the paper, but their requested experiments were run. Xnbz never replied to rebuttal (main objection is lack of real world application of assumptions). The last response from authors, post-revisions, included more semi-realistic experiments; although the modalities added are limited, these seem satisfactory to me. (No reviewers responded to the last experiments as they were added late, but I believe they expand the realistic experiment basis sufficiently.)

---

### Decision · Program_Chairs · 2025-01-22

Accept (Poster)